# A Set of Fluorescent Protein-Based Markers for Major Vesicle Coat Proteins in Yeast

**DOI:** 10.3390/membranes15070209

**Published:** 2025-07-13

**Authors:** Xue-Fei Cui, Zheng-Tan Zhang, Jing Zhu, Li Cui, Zhiping Xie

**Affiliations:** State Key Laboratory of Microbial Metabolism, Ministry of Education Key Laboratory for the Genetics of Developmental and Neuropsychiatric Disorders, and Joint International Research Laboratory of Metabolic & Developmental Sciences, School of Life Sciences and Biotechnology, Shanghai Jiao Tong University, Shanghai 200240, China

**Keywords:** transport vesicles, fluorescent markers, COPII, COPI, AP coats, retromer, yeast

## Abstract

In eukaryotic cells, vesicle-mediated transport interconnects the endomembrane system. These vesicles are formed by coat proteins via deformation of donor membranes. Here, we constructed a set of fluorescent protein-based markers for major coat protein complexes in the yeast model system, and examined their subcellular localization patterns. Our markers covered COPII, COPI, AP-1, AP-2, AP-3, and retromer complexes. Our live cell imaging demonstrates that COPII puncta were primarily associated with the endoplasmic reticulum (ER), and occasionally with early Golgi. COPI was present on both early Golgi and late Golgi/early endosomes. AP-1 puncta were present on late Golgi/early endosomes. AP-2 was present on plasma membrane (PM)-associated puncta, and around the bud neck. AP-3 puncta were present on late Golgi/early endosomes and on the surface of vacuoles. Retromer was present on the surface of vacuoles, late endosomes, and other perivacuolar puncta. Notably, more than half of AP-1 puncta and AP-3 puncta were not associated with the donor compartments where they are thought to be generated, implying that these were coated transport vesicles. This work provides a convenient tool set for the investigation of vesicular transport in yeast and live cell imaging evidence for the presence of certain coated transport vesicles.

## 1. Introduction

Vesicular transport delivers proteins and lipids between membrane structures in eukaryotic cells. These vesicles are formed on the surface of their respective donor compartment through coat protein-mediated membrane deformation, and are generally named after the corresponding coat protein complexes [1,2,3]. Coat proteins are recruited to the donor compartment by upstream signals such as activated Arf family GTPases or phosphoinositides. Coat proteins are released from vesicle surfaces for subsequent reuse before vesicles fuse with acceptor compartments.

The presence of coated vesicles is an ancestral trait shared by all eukaryotes. As in many cell biology studies, yeast *Saccharomyces cerevisiae* is an important model system in the investigation of coat proteins. Four common types of vesicle coats are present in the yeast proteome: coat protein II (COPII), coat protein I (COPI), adaptor protein (AP)/clathrin complexes, and retromer [4,5]. COPII is important for anterograde transport from the endoplasmic reticulum (ER) to the Golgi [1,6]. COPI participates in intra-Golgi transport and retrograde transport from Golgi to ER [6,7]. APs are involved in several transport pathways among Golgi/endosomes, vacuoles, and plasma membrane (PM) [2,5]. Although APs are sometimes referred to as clathrin coats, clathrin is not an essential component in all AP coats. Yeast has three AP complexes, AP-1, AP-2, and AP-3, in which only AP-1 and AP-2 function with clathrin. Retromer mediates the formation of tubular vesicles that transport cargos from endosomes to Golgi or PM [3,8].

Although many vesicle coats have been discovered for more than thirty years, there remain many unresolved questions regarding how and where they function, for which imaging data can provide critical clues. In particular, despite yeast being a classical model for organelle studies, even concepts such as the identities of organelles are subject to update in light of live cell imaging data from others and us [9,10]. Therefore, we believe that a systematic re-evaluation of coat protein subcellular localization is warranted. For this purpose, here we constructed a set of green and red fluorescent protein-based fusion constructs for six common coat protein complexes in yeast, including COPII, COPI, AP-1, AP-2, AP-3, and retromer. We then characterized the subcellular localization of the six coat protein complexes by referencing a validated set of organelle markers. This work provides a set of imaging data obtained under a uniform experimental condition that can aid the systematic understanding of coat functions, and a convenient tool set for coat protein visualization in future yeast cell biology studies.

## 2. Materials and Methods

### 2.1. Yeast Culturing and Fluorescence Microscopy

Yeast cells were first inoculated into YPD (1% yeast extract, 2% peptone, 2% glucose) liquid medium on the first day. On the morning of the second day, cells were diluted to OD_600_ = 0.2 and cultured until the OD_600_ reached approximately 0.6 to 0.8 (i.e., mid-log phase) for microscopy observation.

Glass slides were coated with 1 mg/mL Concanavalin A. Next, 2 OD_600_ of yeast cells was collected by centrifugation, washed in water, and suspended in 300 μL water (note that suspension in water can lead to hypoosmotic shock). Then, 80 μL of suspended cells was allowed to precipitate on the slide for 3 min. Image z-stacks (15 slices, 0.5 μm step size) were collected on an Olympus IX83 inverted fluorescence microscope(Olympus, Tokyo, Japan) equipped with a Photometrics Prime BSI camera (Photometrics, Tucson, USA). The objective lens used was UPLSAPO100XO (100×/1.5). For fluorescent image captures, excitation intensity was set to 100%, and exposure time for each frame was 100 ms for GFP channel or 200 ms for RFP channel.

Note that because coat constructs were expressed at endogenous levels in this work, a wide field fluorescent microscope equipped with a sCMOS camera was chosen over laser scanning confocal microscopes for better detection of weak fluorescent signals [11].

### 2.2. Yeast Strain Construction

Strains expressing fluorescent protein constructs were generated by transformation of linearized integration plasmids into parental strains. To visualize individual coat protein subunits, plasmids encoding tagged coat proteins were introduced into the DJ01 strain (BY4741 *trp1Δ*: natMX) [9]. For colocalization analysis, a second round of transformation was performed to introduce the additional constructs. Plasmids used in this work and the restriction sites used for plasmid linearization are listed in Appendix A. Strains used in this work are listed in Appendix A.

For transformation, parental strains were cultured to the mid-log phase. Then, 2 OD_600_ of yeast cells was collected by centrifugation and washed by LiTE buffer (100 mM lithium acetate, 10 mM Tris, 1 mM EDTA, pH 7.5). Yeast cells were resuspended in 320 μL LiTE/PEG (35% PEG, 100 mM lithium acetate, 10 mM Tris, 1 mM EDTA, pH 7.5), together with 5 μL of linearized plasmid and 10 μL of ssDNA (5 mg/mL, Sigma-Aldrich, Darmstadt, Germany). The suspension was heat shocked at 42 °C for 50 min. Cells were collected by centrifugation and washed with LiTE working buffer. Finally, cells were resuspended with sterile water and spread on selective synthetic medium agar plates. The synthetic media were based on SMD (0.67% yeast nitrogen base without amino acids, 2% glucose, adenine 30 mg/L, histidine 20 mg/L, leucine 50 mg/L, tryptophan 50 mg/L, lysine 30 mg/L, uracil 20 mg/L, methionine 30 mg/L, 2% agar), and lacked either uracil or tryptophan as needed for appropriate auxotrophic marker selection. Following incubation at 30 °C for 3 days, individual colonies were streaked onto fresh selective medium plates.

### 2.3. Plasmid Construction

For this work, the following plasmids were used: those encoding tagged coat proteins, those encoding organelle or plasma membrane markers, and those encoding markers for coat functionality tests (Appendix A). Plasmids encoding six organelle markers were from a previous publication [9]. Plasmids encoding tagged coat proteins, a plasma membrane marker, and tagged Vps10 and Sna2 were newly constructed as detailed in Appendix A. The overall schemes of their construction are provided below.

#### 2.3.1. GFP- or 2mCherry-Tagged Coat Proteins

This set of plasmids is based on two backbone plasmids, CLHN-GFP-URA (*K.l.*) [12] and CLHN-2ycomCherry-TRP (*K.l.*). CLHN-2ycomCherry-TRP was constructed from CLHN-mCherry-TRP (*K.l.*) [9] by replacing the original mCherry ORF with two copies of yeast codon optimized mCherry ORF amplified from P_ATG8_-ycomCherry-Atg8- NatMX6 [13].

To construct plasmids encoding tagged coat subunits, ORFs of the coat protein genes were amplified from yeast genomic DNA via PCR. Amplified coat ORFs and backbone plasmids were digested by restriction enzymes, and ligated via a recombination kit (ClonExpress II One Step Cloning Kit, Vazyme, Nanjing, China).

#### 2.3.2. 2GFP-Tagged AP-2 Subunits

To improve the signal levels of AP-2 subunits, Apl1 and Aps2, plasmids encoding dual GFP-tagged versions were constructed by inserting amplified coat ORFs into KT209-2GFP [9].

#### 2.3.3. mNeonGreen-Tagged COPII Subunits

Plasmid Cl209 was generated by replacing the *URA3* (*K.l.*) cassette of CLHN-GFP-URA (*K.l.*) with *URA3* (*C.a.*) from KT209-2GFP. Fragments containing promoters and ORFs of Sec13 and Sec23 were amplified from genomic DNA, and inserted into Cl209 to generate P*_SEC13_*-Sec13-Cl209 and P*_SEC23_*-Sec23-Cl209. ORF of mNeonGreen was amplified from CLHN-mNeonGreen-URA [14] and inserted into these two plasmids to generate P*_SEC13_*-Sec13-mNG-Cl209 and P*_SEC23_*-Sec23-mNG -Cl209.

#### 2.3.4. mCherry-Tagged Sso2 (Plasma Membrane Marker)

A fragment containing promoter and ORF of Sso2 was amplified from genomic DNA, and inserted into CLHN-mCherry-TRP (*K.l.*) to generate P*_SSO2_*-mCherry-Sso2-TRP (*K.l.*).

#### 2.3.5. GFP- or mCherry-Tagged Vps10

Plasmids Vps10-GFP-URA (*K.l.*) and Vps10-2ycomCherry-TRP (*K.l.*) were constructed following the method detailed in Section 2.3.1.

Vps10-2ycomCherry-MET15 was generated by replacing the *TRP* (*K.l.*) cassette of Vps10-2ycomCherry-TRP (*K.l.*) with MET15 (*S.c.*) from CLHN-MET15 [15].

#### 2.3.6. GFP- or mCherry-Tagged Sna2

Fragments containing promoter or ORF of Sna2 were each amplified from genomic DNA. GFP and mCherry were separately amplified from CLHN-GFP-URA (*K.l.*) and CLHN-ycomCherry-TRP (*K.l.*). The promoter and ORF of Sna2, together with GFP or ycomCherry, were inserted into CLHN-MET15 to generate P*_SNA2_*-GFP-Sna2- MET15 or P*_SNA2_*-ycomCherry-Sna2- MET15.

### 2.4. Western Blot

First, 3 OD_600_ of cells was collected, washed in water, resuspended in 10% (*w*/*v*) trichloroacetic acid (TCA), and incubated on ice for 30 min. The sample was centrifuged at 10,000× *g* for 1 min, resuspended in acetone, centrifuged again and washed with acetone twice. The pellet was air-dried for 1–2 h with tube caps open. Next, 100 μL glass beads and 100 μL lysis buffer (62.5 mM Tris-HCl pH 6.8, 2% SDS, 1 mM EDTA, 6 M urea) were added to each sample. The pellet was mechanically disrupted in a bead mill with three rounds of vortexing at 60 Hz for 5 min each. Lysate was centrifuged at 10,000× *g* for 10 min, and the supernatant was collected. Then, 50 μL of supernatant was mixed with an equal volume of 2× Loading Buffer (62.5 mM Tris-HCl pH 6.8, 2% SDS, 2% glycerol, 4% β-mercaptoethanol, 2% Bromophenol blue), and then denatured at 70 °C for 10 min. Next, 3 μg of protein (quantified by Bradford assay) was loaded per sample for SDS-PAGE and immunoblotting analysis. Antibodies used were anti-GFP (AB0005, Abways, Shanghai, China), anti-mCherry (LF312S, EpiZyme, Shanghai, China), anti-GAPDH (AB2000, Abways).

### 2.5. Spot Assay

Yeast strains were inoculated in YPD liquid medium and incubated overnight at 23 °C. Overnight cultures were diluted to OD_600_ = 0.2 and cultured for an additional 2.5 h, with temperature-sensitive (TS) mutants maintained at 25 °C while other strains were cultured at 30 °C.

For each sample, 1 OD_600_ of cells was harvested, and adjusted to have an optical density of 1 OD_600_., followed by 10-fold serial dilutions in YPD medium. Next, 3 μL aliquots of each dilution were spotted onto YPD agar plates and then incubated at 25 °C and 37 °C for 2–3 days.

### 2.6. Mating Assay

Yeast cells were cultured as described in Section 2.1. Next, 2 mL of experimental strain culture was mixed with 1 mL of BY4742 [16] strain culture. The mixed culture was then incubated at 30 °C for 2 h, and analyzed by microscopy.

## 3. Results

### 3.1. Visualization of All Six Coat Protein Complexes

To visualize the six coat protein complexes in live yeast cells, we picked two subunits from each coat complex, and prepared plasmid vectors designed to integrate into the genome to express fluorescent protein chimeras under native promoters. For each coat subunit, we prepared two constructs, one for a GFP or mNeonGreen fusion and one for tandem double mCherry (2mCherry) fusion, resulting in a total of twenty-four coat subunit constructs (Table 1). We chose tandem mCherry to improve the signal intensity of the red constructs. Under our hardware setup, the signal strength of 2mCherry constructs was still weaker than that of the GFP constructs (Figure 1). As a result, we generally saw more weakly fluorescent puncta with GFP constructs than with the corresponding mCherry constructs. Immunoblotting analysis demonstrated that all coat constructs were expressed with expected molecular weights (Figure A1).

We first examined the tagged coat subunits individually (Figure 1) (see also Figure A2 for fluorescent images presented in inverted grey scale mode). When using the same exposure setting (i.e., the same combination of excitation light intensity and exposure time), the resulting signal strength of the coat proteins could be classified into three groups. The first group, containing subunits of COPII and COPI, was the brightest. The second group, containing most other coat proteins, displayed intermediate levels of fluorescent signal. The last group, containing subunits of AP-2 (Apl1 and Aps2), was the weakest. We therefore introduced tandem double GFP (2GFP) tags for AP-2 subunits. With 2GFP tagging, the signal of Apl1 became close to the level of the intermediate group, although that of Aps2 was still noticeably weaker.

To obtain a panoramic view of the subcellular distribution of coat proteins, we collected z-stacks covering the entire depth of yeast cells. For presentation, we selected two slices, one at the top of cells to focus on structures at or close to the PM, and one in the middle showing a cross-sectional view of the internals. We also included a z-stack projection to display all the visible fluorescent structures. Overall, all tagged coat proteins displayed various forms of punctate signal, and subunits of the same coat complex displayed the same localization pattern. Among coat constructs displaying intermediate and weak signal levels, GFP-tagged constructs generally displayed more puncta than the corresponding mCherry constructs, which is expected given that the red constructs were dimmer. For two coats, COPII and AP-2, we noticed many fluorescent puncta close to the PM (visible in the top slice, and in cell periphery in the center slice), which likely reflect their association with either PM associated ER (for COPII) or the PM (for AP-2). In the case of AP-2, because of the overall weak signal, only green constructs displayed visible puncta. Besides PM proximal small puncta, AP-2 was additionally heavily concentrated around bud necks. Some COPII and most retromer puncta were concentrated along intracellular spheres, presumably the nuclear ER (for COPII) and vacuoles (for retromer). Retromer also dimly decorated the boundary of the intracellular spheres. Compared with puncta of most other coat complexes, COPI puncta were larger, less sharp, and displayed irregular contours.

### 3.2. Functionality of Coat Protein Constructs

Next, we evaluated if each coat chimera was functional. Trafficking pathways mediated by COPII and COPI were essential for survival [17,18,19]. We therefore used vegetative growth as the readout for their functions. As expected, cells carrying temperature sensitive alleles of *sec13* (COPII), *sec23* (COPII), *cop1* (COPI), and *sec21* (COPI) were unable to grow at the non-permissive temperature of 37 °C (Figure 2). In contract, mutant cells complemented with a copy of mNeonGreen-tagged version of coat subunits and cells expressing tagged version of coat subunits as the sole copy were able to grow. Growth of Sec23-mNeonGreen expressing cells was slightly slower than normal, indicating that this construct was partially functional. The other COPII and COPI constructs were all functional.

AP-2 has an important role in regulating polarized growth signaling [20]. When cells from a-type experimental strain were incubated in the presence alpha-type cells (BY4742), AP-2 knockout cells displayed higher incidence of dumbbell morphology, which was indicative of abnormal formation of mating projection (Figure 3A,B). Cells expressing tagged versions of AP-2 subunits as the sole copies displayed morphology indistinguishable from wild-type cells, suggesting that the AP-2 chimeras were functional.

Both AP-1 and AP-3 are important for the subcellular trafficking of Sna2 [21]. Sna2 is an integral membrane protein normally concentrated on vacuolar membrane. It contains two sorting motifs, each recognized by AP-1 or AP-3. In AP-1 defective cells (*apl2Δ* and *apm1Δ*), Sna2 was partially relocated from the vacuolar membrane to the plasma membrane (Figure 4). In AP-3 defective cells (*apl5Δ* and *apm3Δ*), Sna2 became heavily concentrated on the plasma membrane with limited intracellular presence. In cells expressing tagged AP-1 and AP-3 subunits, Sna2 was primarily localized to the vacuolar membrane as in wild-type cells, indicating that the AP-1 and AP-3 subunit chimeras were functional.

Retromer mediates the intracellular sorting of Vps10 [22]. In wild-type cells, Vps10 was present on multiple intracellular puncta (Figure 5). In retromer mutants, Vps10 was mis-sorted to the vacuolar membrane (note that vacuoles were additionally fragmented in vps5D cells, which is a known phenotype). In cells expressing tagged retromer subunits, Vps10 was distributed on intracellular puncta as in wild-type cells, indicating that the retromer subunit chimeras were functional.

### 3.3. Co-Localization Between Subunits of the Same Coat Complex

Since we have a set of two subunits by two color tagging for each coat complex, we verified the proper integration of fusion proteins into coat complexes by co-localization analysis (Figure 6). The analysis was performed in the following fashion: for a coat with subunit x and y, we constructed two strains, one co-expressing green x with red y, and one co-expressing red x with green y. For all twelve pairs of co-expressed green and red coat constructs, we observed clear colocalization between the green puncta and red puncta, implying that the tagged coat subunits were complexed with the other subunits of the coat complex they belong to.

### 3.4. Identification of the Subcellular Localizations of the Coat Complexes by Co-Localization Analysis with Organelle Markers

Subcellular localization of coat proteins provides important clues to their function. In principle, coat protein may be observed at the donor compartment prior to or during vesicle budding, on transport vesicles, or at the acceptor compartment if uncoating is not yet finished. To identify the subcellular localization of coat complexes, we co-expressed green fluorescent protein-tagged coat subunits with red fluorescent protein-tagged location markers. The following markers were used: mCherry-HDEL (ER), Anp1-mCherry (early Golgi), Sec7-DuDre (late Golgi/early endosomes), Vps4-DuDre (late endosomes), Vph1-mCherry (vacuoles), mCherry-Sso2 (PM), and Cox4-DuDre (mitochondria).

In live cell imaging, we found that subunits of COPII, Sec13 and Sec23 co-localized frequently with ER marker mCherry-HDEL (Figure 7 and Figure 8). In yeast, two prominent sections of ER are visible, one along the nucleus, one close to PM. COPII co-localized with both sections. Close to half of all COPII puncta could be seen along ER structures. Since the membrane network of ER is easier to observe when in the middle sections, and becomes fuzzy in top and bottom sections, it was likely that our colocalization analysis missed some ER structures in top and bottom sections, and that the colocalization ratio represents an underestimation. It is also worth noting that the peripheral ER network is associated with PM via ER-PM contact sites; thus under our experimental condition, the resolution of wide field microscopy was insufficient to resolve the peripheral ER from PM. As a result, we could technically observe a high ratio of colocalization between coat subunits and PM marker mCherry-Sso2. We also observed low ratios of colocalization between COPII and early Golgi marker Anp1-mCherry (0.03 colocalization/GFP or 0.12 colocalization/RFP). Despite the low values, they were substantially higher than the ratios with late Golgi/early endosomes, late endosomes, and mitochondria, thus representing true colocalization instead of random overlap.

Subunits of COPI, Cop1 and Sec21, co-localized with markers of two organelles, early and late Golgi (Anp1-mCherry and Sec7-DuDre) (Figure 9 and Figure 10). Unlike COPII subunits that mostly manifested as sharp puncta, COPI subunits manifested as irregular shapes that were difficult to separate from each other. Therefore, when quantifying colocalization, we did not count individual puncta and instead used Pearson’s correlation between images of the two fluorescent channels. Correlation values around 0.4 were observed with most organelle markers, indicating that it represents the background value under our experimental condition. In contrast, correlations at about 0.6–0.7 were observed between COPI subunits and markers of early Golgi and late Golgi/early endosomes (Anp1-mCherry and Sec7-Dudre).

Subunits of AP-1, Apl2 and Apm1, co-localized with Sec7-DuDre, the marker for late Golgi/early endosomes (Figure 11 and Figure 12). The observed colocalization ratios were about 0.3–0.5 colocalization/GFP, or about 0.2 colocalization/RFP.

As mentioned, when observing coat proteins individually, subunits of AP-2, Apl1 and Aps2 reside on the bud neck and multiple additional puncta. These additional puncta co-localized with mCherry-Sso2, the marker for PM (Figure 13 and Figure 14). The observed colocalization ratio was about 0.6–0.8 colocalization/GFP. For spherical structures such as the PM, nuclear ER, and vacuoles, their signal was easier to see in the middle sections, and essentially non-recognizable in top and bottom sections. Thus, the quantified colocalization ratio here was an underestimation. AP-2 subunits also co-localized with peripheral ER labeled by mCherry-HDEL, as a result of the resolution limit mentioned early.

Subunits of AP-3, Apl5 and Apm3, co-localized with two location markers, Sec7-DuDre for late Golgi/early endosomes and Vph1-mCherry for vacuoles (Figure 15 and Figure 16). Notably, AP-3 subunits only displayed punctate distribution. AP-3 puncta could be seen co-localizing with Sec7-positive Golgi puncta (0.10–0.14 colocalization/GFP, or 0.11–0.15 colocalization/RFP), or residing along the vacuolar membrane marked by Vph1 (0.13–0.17 colocalization/GFP). AP-3 did not co-localize with late endosome marker Vps4, despite the latter also residing close to the vacuolar membrane.

Subunits of retromer, Vps5 and Vps26, also co-localized with two location markers, Vps4-DuDre for late endosomes and Vph1-mCherry for vacuoles (Figure 17 and Figure 18). As mentioned previously, retromer subunits appeared to reside on the boundary of intracellular spheres and several spots along the boundary. Co-localization data confirmed that the spheres were vacuoles marked by Vph1. In the case of the puncta, we found that their colocalization ratios with late endosomes were 0.2–0.4 colocalization/GFP or 0.3–0.4 colocalization/RFP. Their colocalization ratios with vacuoles were 0.3–0.4 colocalization/GFP, which represented an underestimation resulting from the spherical structure of vacuoles.

## 4. Discussion

In the present work, we constructed plasmids for the tagging of coat proteins with green and red fluorescent proteins. Our constructs covered all six major coat complexes in yeast, including COPII, COPI, AP-1, AP-2, AP-3, and retromer. Via co-localization analysis, we found that COPII was mainly present in ER-associated puncta and occasionally on early Golgi. COPI was present on early Golgi and late Golgi. AP-1 was present on late Golgi. AP-2 was present in PM-associated puncta and the bud neck. AP-3 was present on late Golgi, vacuole-associated puncta, and other cytoplasmic puncta. Retromer was present in three populations, one in puncta form on late endosomes, one in puncta form but not on late endosomes, and one spread out on the vacuolar membrane.

COPII is essential in the anterograde trafficking from the ER to Golgi [1,6]. Our observation of COPII presence on the ER and early Golgi is consistent with the established role of COPII. COPII association with early Golgi is potentially transient [23], and may represent the docking stage in the life cycle of transport vesicles before uncoating and fusion.

COPI is thought to mediate intra-Golgi transport and retrograde transport from Golgi back to the ER [6,7,24,25]. It has been shown to co-localize with early or late Golgi markers in yeast live cell imaging [25,26]. Since we observed the presence of COPI on both early Golgi and late Golgi, but not on the ER, the COPI puncta we observed likely represent the sites of COPI vesicle budding on donor compartments instead of docking sites on acceptor compartments.

AP-1 mediates anterograde traffic from Golgi to endosomes or PM, and possibly intra-Golgi transport [10,27,28]. We observed the presence of AP-1 on late Golgi/early endosomes, but not late endosomes. This is consistent with published yeast imaging studies [26,27] and confirms that late Golgi/early endosomes are the site of AP-1 vesicle budding.

AP-2 mediates endocytosis [29,30]. AP-2-positive PM-associated puncta thus likely represent sites of endocytosis [30,31]. Whether the population of AP-2 concentrated at bud necks participates in endocytosis is not known [29,32].

AP-3 mediates the trafficking of a group of proteins from late Golgi/early endosomes to the vacuole [10,33]. Consistent with previous reports [10,33,34,35], we observed AP-3 puncta on late Golgi/early endosomes and on the surface of vacuoles. Considering that late Golgi/early endosomes do not strongly associate with vacuoles [9,10], the AP-3 puncta on the surface of vacuoles we observed likely represent a different population from those at late Golgi/early endosomes. These peri-vacuolar AP-3 puncta may represent docked AP-3 vesicles before fusion [34]. The absence of AP-3 on late endosomes is consistent with the general understanding that the AP-3 pathway does not transit through late endosomes.

Retromer is thought to mediate the recycling of proteins from the endosomes back to Golgi or PM [3,22]. Under our experimental condition, we observed essentially perfect co-localization between Vps5 and Vps26, each belonging to the Vps5-Vps17 subcomplex and Vps26-Vps29-Vps35 subcomplex, consistent with the reported strong association between the two subcomplexes in yeast [8]. Our observation of retromer presence on late endosomes and vacuoles, but not late Golgi/early endosomes, supports the hypothesis that late endosomes and vacuoles are sites of retromer function [36,37,38]. The Vps26-Vps29-Vps35 subcomplex is known to form a complex with Atg18 on the vacuolar surface [36,37]. The relationship between Vps5-Vps17 subcomplex and Atg18 remains to be fully explored.

An often debated topic about coat proteins is whether the puncta formed by them represents coated vesicles. Assuming that our current understanding of the beginning and ending of individual traffic route is correct, if a punctum is clearly away from the donor compartment, it becomes likely that the punctum is a coated vesicle. On the other hand, if a punctum is associated with its donor compartment, it may represent either a forming vesicle or a site of vesicle formation, and drawing a definitive conclusion will require additional evidence. We observed that about half of COPII puncta were associated with ER. For the remaining half of COPII puncta, as discussed previously, some may be still associated with ER structures that are on top and bottom sections, which are difficult to recognize. Thus, we have not yet reached a reliable estimation on the actual ratio of transport vesicles among the remaining half of COPII puncta. This type of underestimation constraint also applies to AP-2 and retromer, in which the difficulty of observing PM and vacuoles at top/bottom slices prevents us from concluding the ratios of transport vesicles. However, in the case of AP-2, given the already high ratios of observed colocalization with PM, we suspect that essentially all AP-2 puncta are PM-associated. For COPI, our imaging data indicate that they are strongly associated with the Golgi. For AP-1, because only less than half of them are positive for late Golgi/early endosome, and we are not constrained by the underestimation issue, we conclude that about half of AP-1 puncta represent transport vesicles away from their donor compartment. For AP-3, clearly the majority of the puncta are not associated with its donor compartment, late Golgi/early endosomes, and thus likely are transport vesicles.

In summary, judging from the localization patterns of individual coat subunits, the colocalization between subunits of the same coat complexes, and the distribution of coat subunit in relation to organelle markers, we conclude that our set of fluorescent protein fusion constructs labeled all six coat complexes correctly. Our co-localization data are overall consistent with the current understanding of the coat complexes, but also highlight several cases where the functional significance of the observed localization sites awaits future investigations (Table 2).

Our coat tagging plasmids constitute a convenient tool set to observe coat protein localization in yeast cells with a strain background and genotype of choice. This is different from the existing proteome scale GFP-tagging library, which was constructed with a specific strain background [39,40,41,42]. The GFP-tagging library excels at providing a panorama of the whole proteome under various conditions [42]. However, adapting the library to other strain backgrounds is technically complicated. In addition, owning to the nature of large-scale work, not all ORFs were covered by the library, including certain coat proteins. For research projects with coat proteins in focus, our tool set is a better fit because it provides better coverage, offers both green and red colors, and is backed by functional validation data.

## Figures and Tables

**Figure 1 membranes-15-00209-f001:**
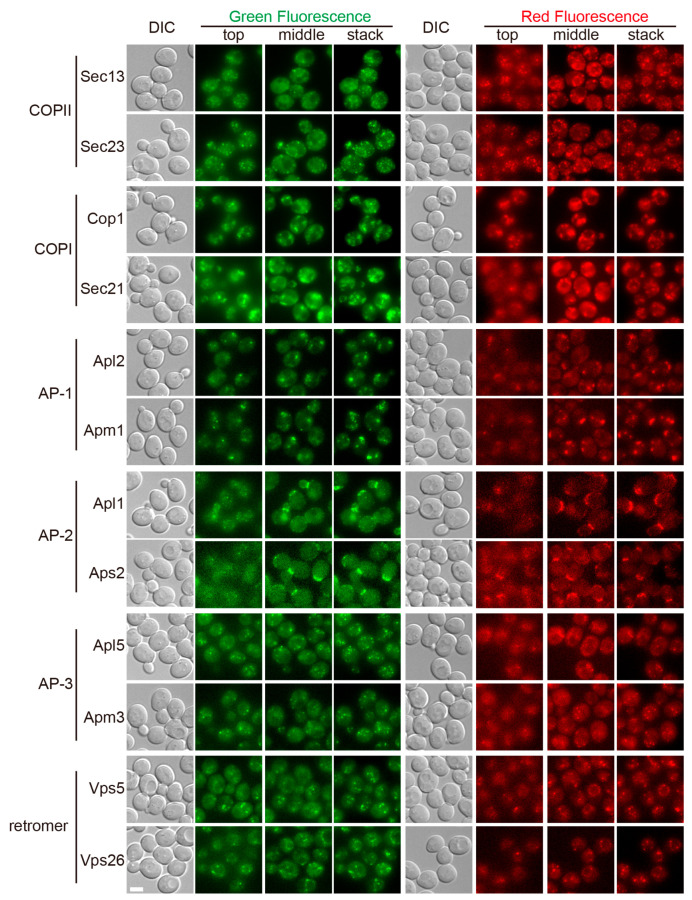
Visualization of individual coat subunits. Yeast cells expressing the indicated coat subunits tagged with green or red fluorescent proteins were cultured to mid-log phase and imaged. For the fluorescent channel from z-stacks covering the entire depth of yeast cells, top and middle sections and max-intensity projections of the whole stacks are presented. DIC, differential interference contrast. Scale bar, 3 μm.

**Figure 2 membranes-15-00209-f002:**
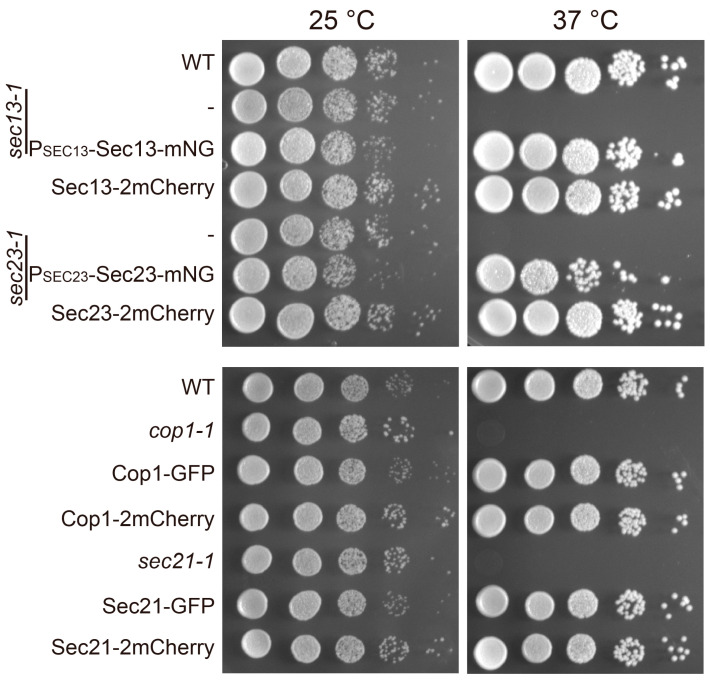
Evaluation of COPII and COPI chimera function by vegetative growth. P_SEC13_-Sec13-mNeonGreen and P_SEC23_-Sec23-mNeonGreen were expressed in the presence of a corresponding untagged temperature-sensitive mutant allele. Sec13-2mCherry, Sec23-mCherry, Cop1-GFP/mCherry, and Sec21-GFP/mCherry were expressed as the sole copies of corresponding ORFs. In addition, 10× serial dilutions were spotted on YPD plates and incubated at the indicated temperatures for 2–3 days.

**Figure 3 membranes-15-00209-f003:**
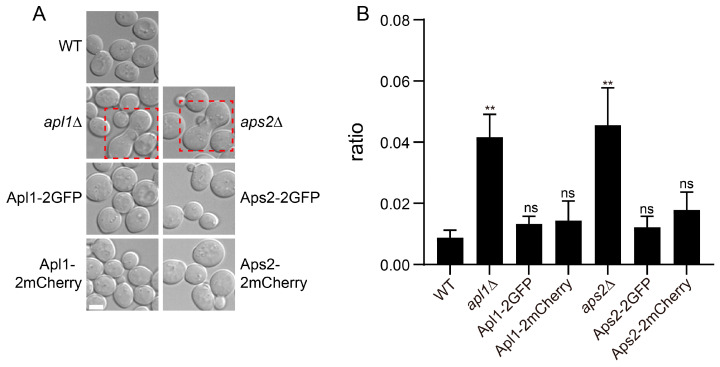
Evaluation of AP-2 chimera function by pheromone-regulated polarized growth. Tagged AP-2 subunits were expressed as the sole copies of corresponding ORFs. Experimental strains, in mating type a, were incubated with alpha-type strains for 2 h at 30 °C and observed. AP-2 mutants displayed higher incidences of dumbbell shaped cells. (**A**) Representative DIC images. Cells with abnormal morphology were indicated by red dashed frames. (**B**) Quantification of the ratios of dumbbell shaped cells. Error bar, standard deviation from three repeats. *t*-test results against wild-type samples were marked as ns, nonsignificant and **, *p* < 0.01.

**Figure 4 membranes-15-00209-f004:**
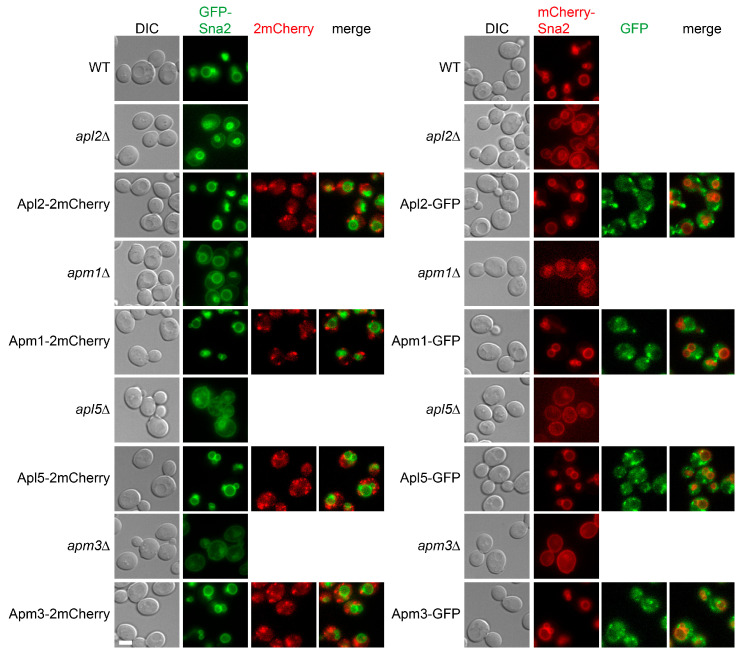
Evaluation of AP-1 and AP-3 chimera function by Sna2 subcellular distribution. GFP or mCherry-tagged Sna2 were used to evaluate functions of mCherry- or GFP-tagged coat subunits, respectively. Tagged coat subunits were expressed as the sole copies of corresponding ORFs. Log phase yeast cells were examined and representative images are shown.

**Figure 5 membranes-15-00209-f005:**
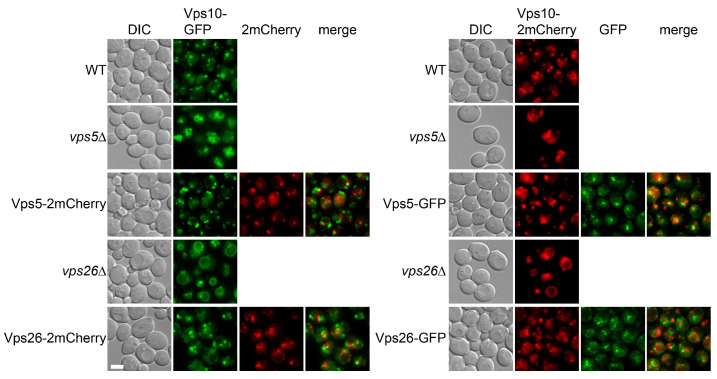
Evaluation of retromer chimera function by Vps10 subcellular distribution. GFP or mCherry-tagged Vps10 were used to evaluate functions of mCherry- or GFP-tagged coat subunits, respectively. Tagged coat subunits were expressed as the sole copies of corresponding ORFs. Log phase yeast cells were examined and representative images are shown.

**Figure 6 membranes-15-00209-f006:**
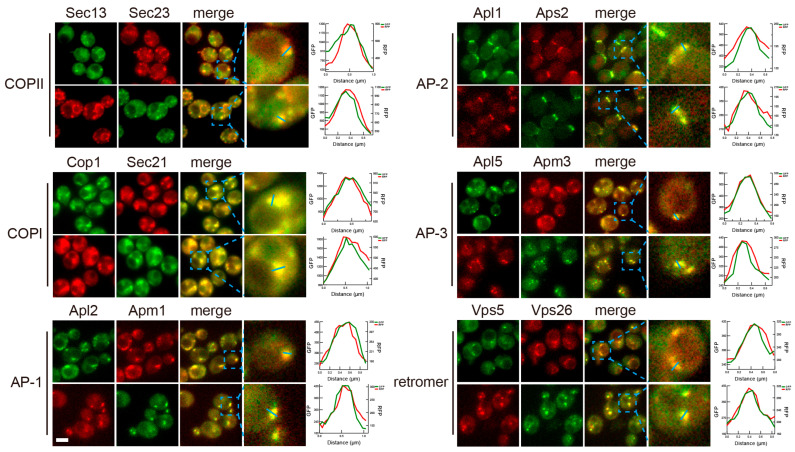
Colocalization between subunits of the same coat complexes. Yeast cells expressing two subunits of the same coat tagged with fluorescent proteins of different color (one green and one red) were cultured to mid-log phase and imaged. Slices of individual fluorescent channels and merged channels are presented. For each merged image, a zoomed in view of an area marked by a blue dashed square is presented in the last image column. Line profiles of fluorescent signal intensities of the two channels along the blue line in the zoomed view are presented to the right, demonstrating colocalization between the two subunits. Scale bar, 3 μm.

**Figure 7 membranes-15-00209-f007:**
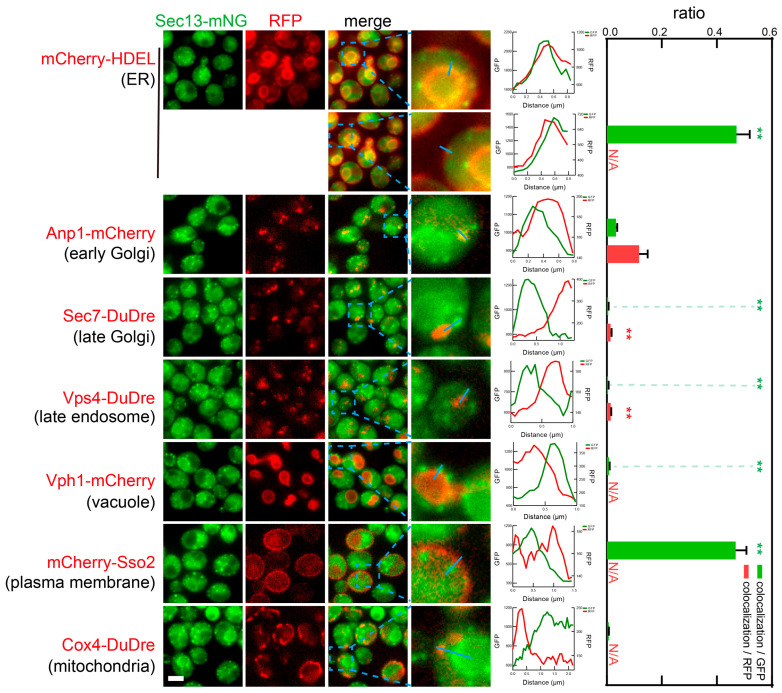
Subcellular localization of COPII subunit Sec13. Yeast cells expressing a mNeonGreen tagged COPII subunit Sec13 together with a red fluorescent protein-tagged location/organelle marker were cultured to mid-log phase and imaged. Slices of individual fluorescent channels and merged channels are presented. For each merged image, a zoomed in view of an area marked by a blue dashed square is presented in the last image column, except for colocalization with red ER marker, two zoomed in areas were presented showcasing peripheral ER and nuclear ER. Line profiles of fluorescent signal intensities of the two channels along the blue line in the zoomed view are presented to the right of the images to demonstrate presence or absence of colocalization. Colocalization ratios are presented as column graphs. Two colocalization ratios are calculated: number of colocalizing puncta/total number of green or red puncta (N/A, not applicable). For structures that do not display punctate signal, including ER, vacuoles, PM, and mitochondria, only one ratio, colocalizing puncta/total number of green puncta, is presented. Scale bar, 3 μm. For ratios of colocalizing puncta/total number of red/green puncta, *t*-test was performed against Anp1-mCherry sample, with significant results marked with red/green stars (** *p* < 0.01).

**Figure 8 membranes-15-00209-f008:**
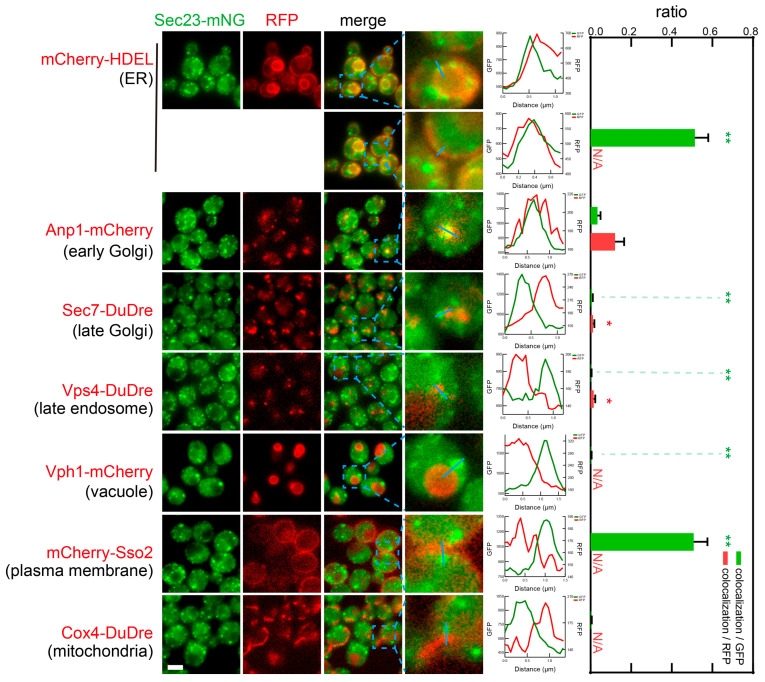
Subcellular localization of COPII subunit Sec23. Yeast cells expressing a mNeonGreen-tagged COPII subunit Sec23 together with a red fluorescent protein-tagged location/organelle marker were cultured to mid-log phase and imaged. Images and quantifications are presented as those in Figure 7. (** *p* < 0.01; * *p* < 0.05).

**Figure 9 membranes-15-00209-f009:**
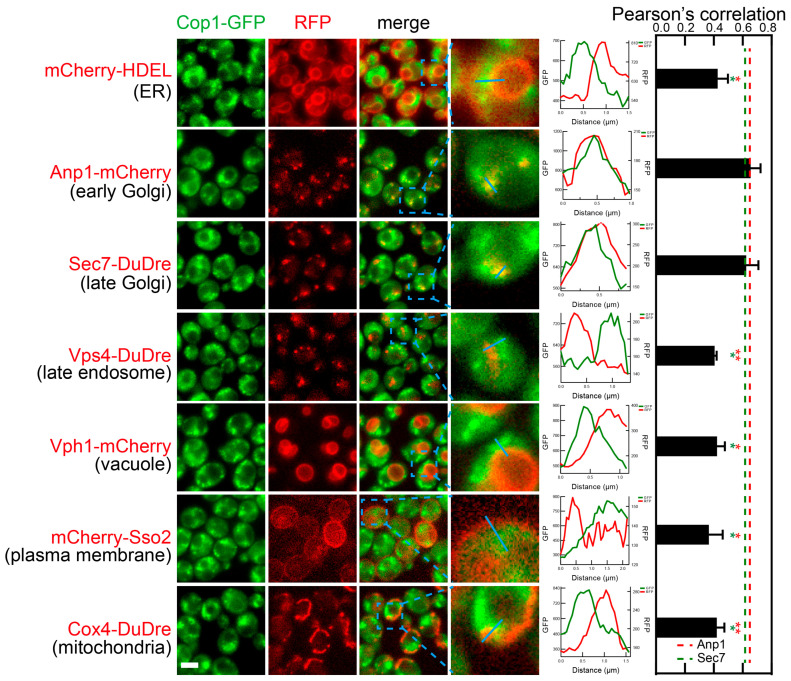
Subcellular localization of COPI subunit Cop1. Yeast cells expressing a GFP-tagged COPI subunit Cop1 together with a red fluorescent protein-tagged location/organelle marker were cultured to mid-log phase and imaged. Images and fluorescent intensity line profiles are presented as those in Figure 7. Because of the irregular often continuous contour, colocalization was quantified via calculating Pearson correlation between two fluorescent channels instead of counting puncta. Error bar, standard deviation. T test was performed against Anp1-mCherry samples (marked with red stars) or sec7-Dudre samples (marked with green stars). (** *p* < 0.01; * *p* < 0.05).

**Figure 10 membranes-15-00209-f010:**
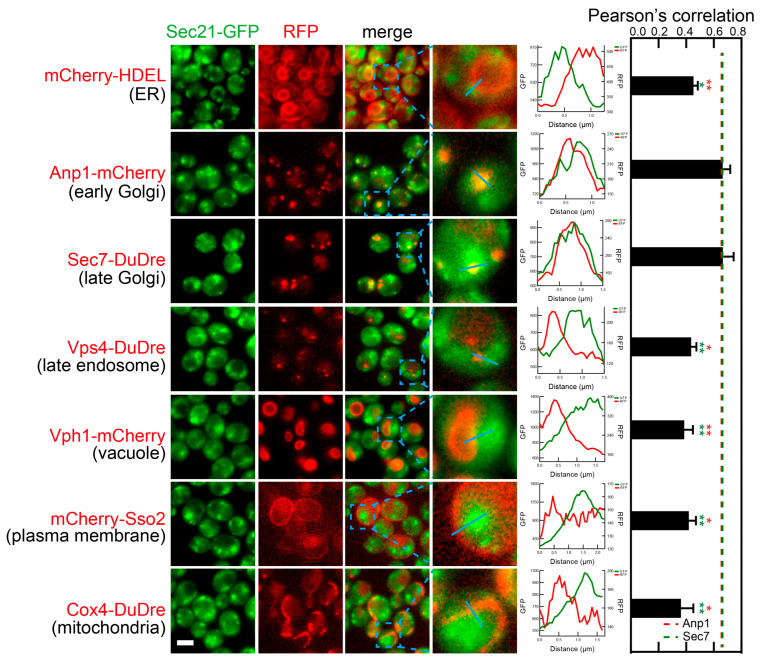
Subcellular localization of COPI subunit Sec21. Yeast cells expressing a GFP-tagged COPI subunit Sec21 together with a red fluorescent protein-tagged location/organelle marker were cultured to mid-log phase and imaged. Images and fluorescent intensity line profiles are presented as those in Figure 7. Because of the irregular often continuous contour, colocalization was quantified via calculating Pearson correlation between two fluorescent channels instead of counting puncta. Error bar, standard deviation. T test was performed against Anp1-mCherry samples (marked with red stars) or sec7-Dudre samples (marked with green stars). (** *p* < 0.01; * *p* < 0.05).

**Figure 11 membranes-15-00209-f011:**
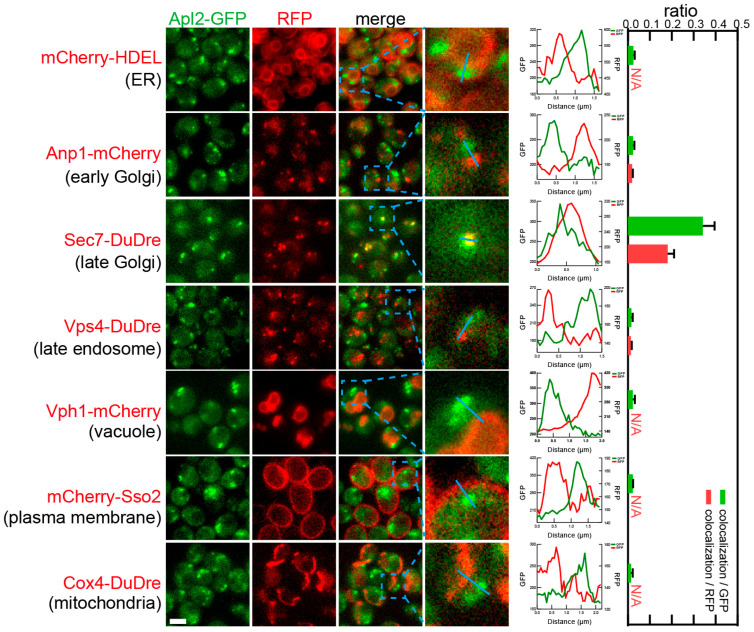
Subcellular localization of AP-1 subunit Apl2. Yeast cells expressing a GFP-tagged AP-1 subunit Apl2 together with a red fluorescent protein-tagged location/organelle marker were cultured to mid-log phase and imaged. Slices of individual fluorescent channels and merged channels are presented. For each merged image, a zoomed in view of an area marked by a blue dashed square is presented in the last image column, except for colocalization with red ER marker, where two zoomed in areas are presented showcasing peripheral ER and nuclear ER. Line profiles of fluorescent signal intensities of the two channels along the blue line in the zoomed view are presented to the right of the images to demonstrate presence or absence of colocalization. Colocalization ratios are presented as column graphs. Two colocalization ratios are calculated: number of colocalizing puncta/total number of green or red puncta. For structures that do not display punctate signals, including ER, vacuoles, PM, and mitochondria, only one ratio, colocalizing puncta/total number of green puncta, is presented. N/A, not applicable. Scale bar, 3 μm.

**Figure 12 membranes-15-00209-f012:**
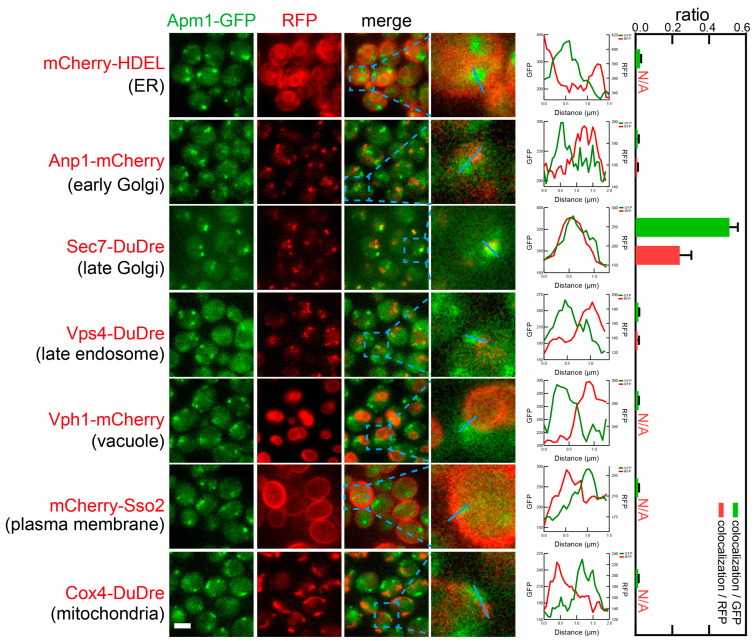
Subcellular localization of AP-1 subunit Apm1. Yeast cells expressing a GFP-tagged AP-1 subunit Apm1) together with a red fluorescent protein-tagged location/organelle marker were cultured to mid-log phase and imaged. Images and quantifications are presented as those in Figure 11.

**Figure 13 membranes-15-00209-f013:**
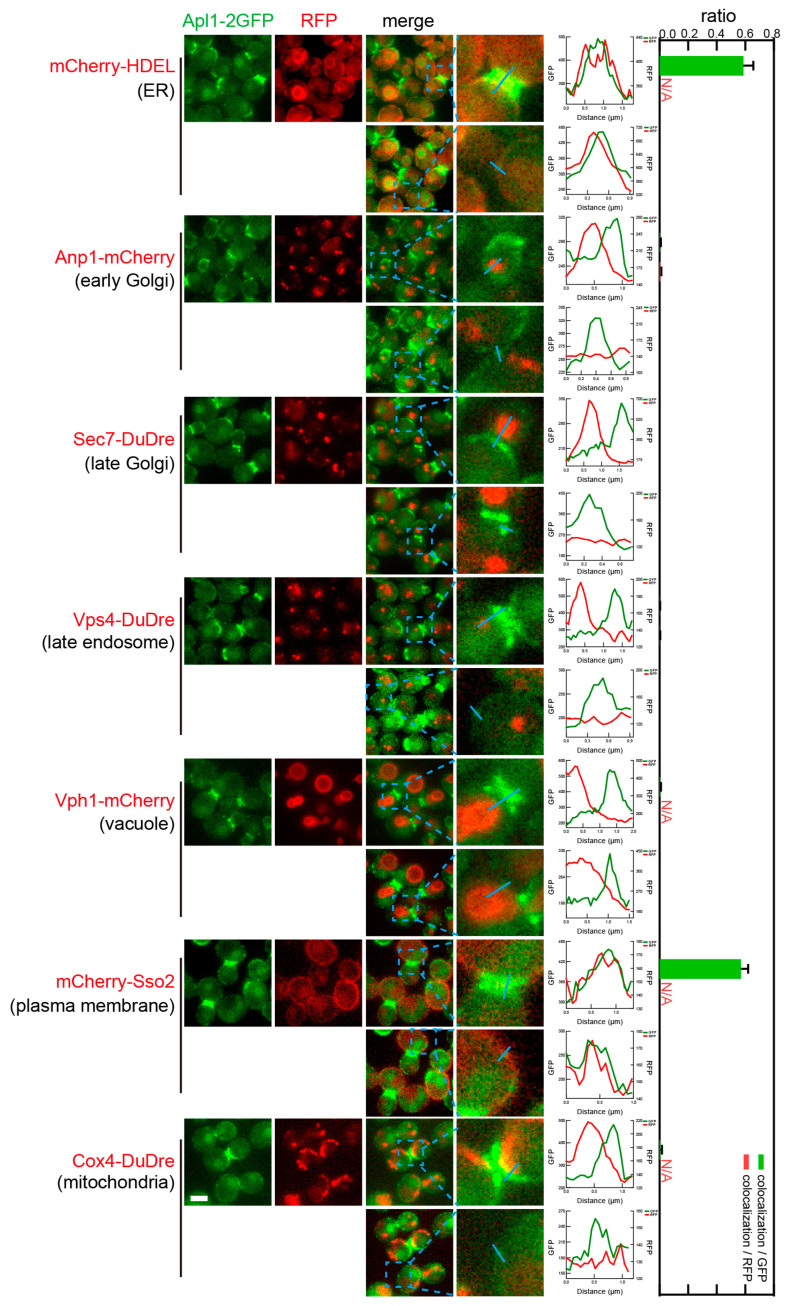
Subcellular localization of AP-2 subunit Apl1. Yeast cells expressing a 2× GFP-tagged AP-2 subunit Apl1 together with a red fluorescent protein-tagged location/organelle marker were cultured to mid-log phase and imaged. Images and quantifications are presented as those in Figure 11, except that two zoomed in areas are analyzed, with the top one focusing on the bud neck, and the lower one focusing on a represented PM segment.

**Figure 14 membranes-15-00209-f014:**
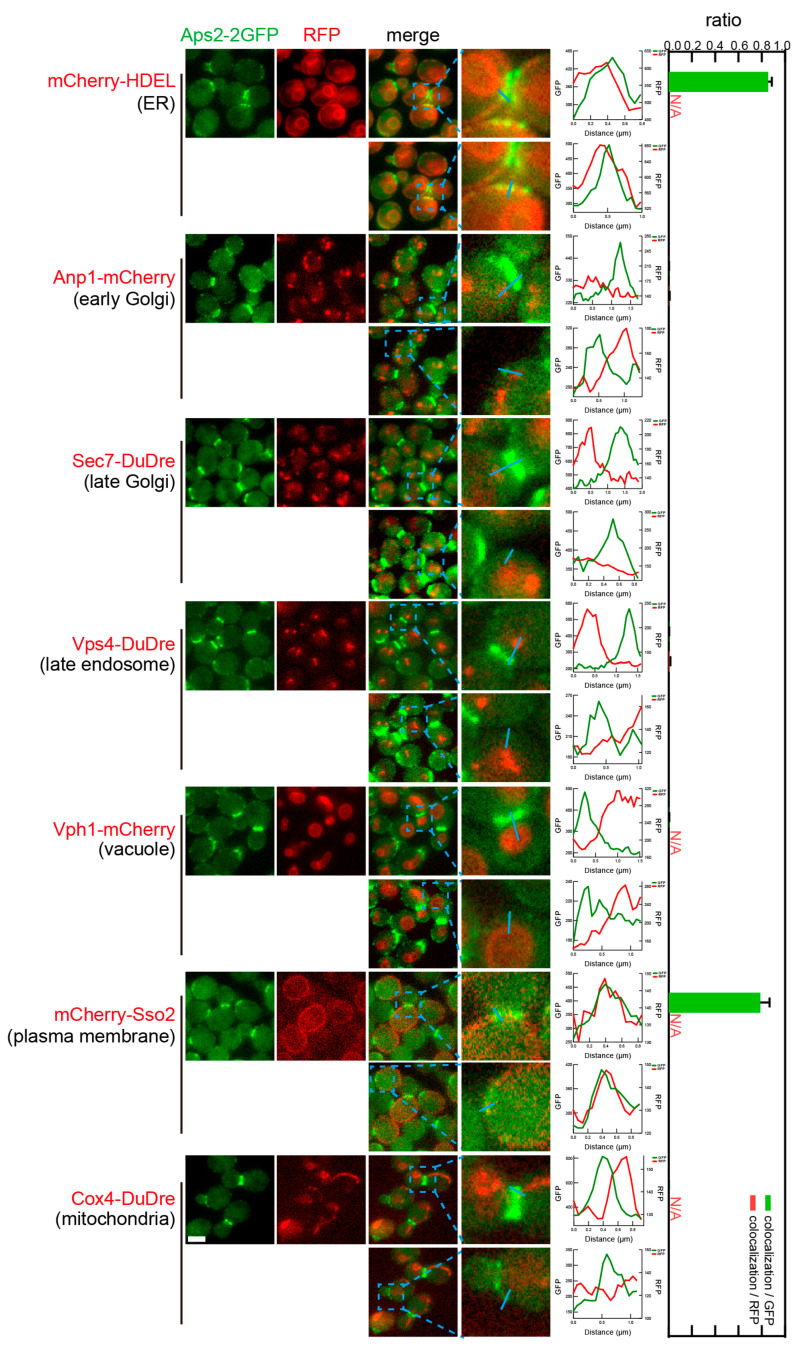
Subcellular localization of AP-2 subunit Aps2. Yeast cells expressing a 2× GFP-tagged AP-2 subunit Aps2 together with a red fluorescent protein-tagged location/organelle marker were cultured to mid-log phase and imaged. Images and quantifications are presented as Figure 13.

**Figure 15 membranes-15-00209-f015:**
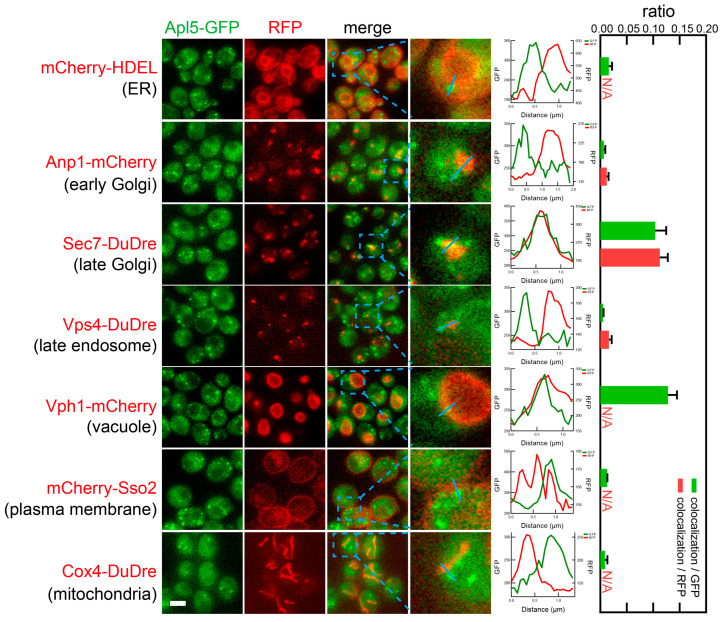
Subcellular localization of AP-3 subunit Apl5. Yeast cells expressing a GFP-tagged AP-3 subunit Apl5 together with a red fluorescent protein-tagged location/organelle marker were cultured to mid-log phase and imaged. Images and quantifications are presented as those in Figure 11.

**Figure 16 membranes-15-00209-f016:**
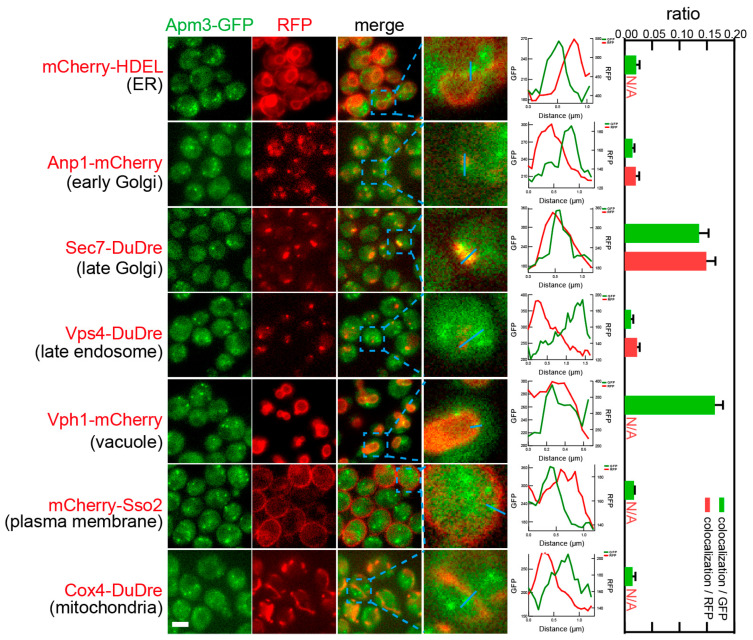
Subcellular localization of AP-3 subunit Apm3. Yeast cells expressing a GFP-tagged AP-3 subunit Apm3 together with a red fluorescent protein-tagged location/organelle marker were cultured to mid-log phase and imaged. Images and quantifications are presented as those in Figure 11.

**Figure 17 membranes-15-00209-f017:**
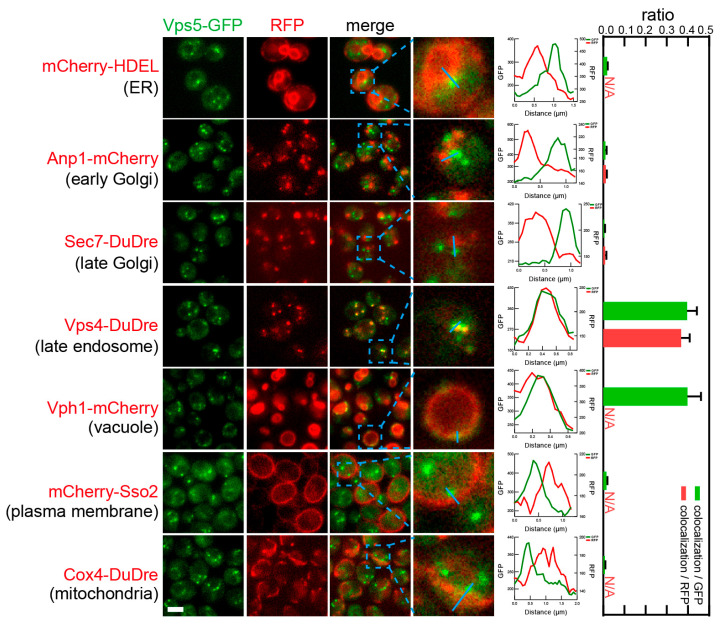
Subcellular localization of retromer subunit Vps5. Yeast cells expressing a GFP-tagged retromer subunit Vps5 together with a red fluorescent protein-tagged location/organelle marker were cultured to mid-log phase and imaged. Images and quantifications are presented as those in Figure 11.

**Figure 18 membranes-15-00209-f018:**
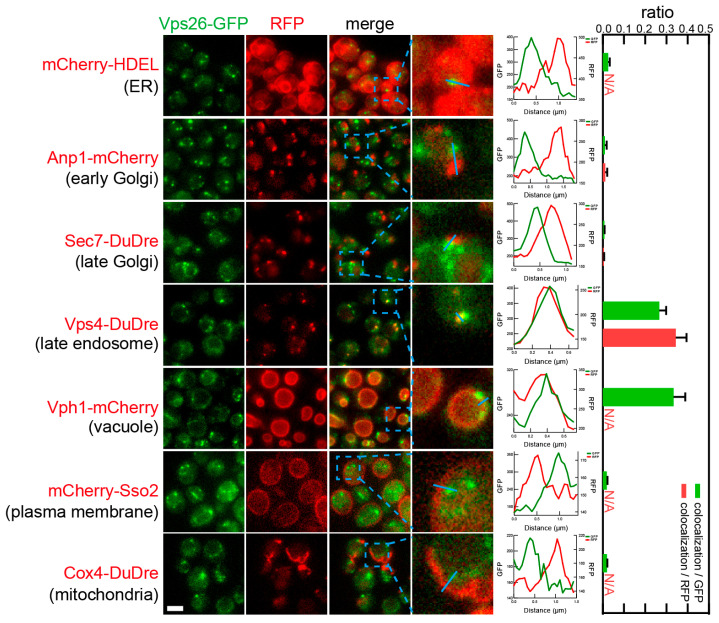
Subcellular localization of retromer subunit Vps26. Yeast cells expressing a GFP-tagged retromer subunit Vps26 together with a red fluorescent protein-tagged location/organelle marker were cultured to mid-log phase and imaged. Images and quantifications are presented as those in Figure 11.

**Table 1 membranes-15-00209-t001:** Coat subunit constructs of coat protein complexes.

Coat	Plasmid	Subunit
COPII	P_SEC13_-Sec13-mNG	Sec13
	Sec13-2mCherry	
	P_SEC23_-Sec23-mNG	Sec23
	Sec23-2mCherry	
COPI	Cop1-GFP	Cop1/α-COP
	Cop1-2mCherry	
	Sec21-GFP	Sec21/γ-COP
	Sec21-2mCherry	
AP-1	Apl2-GFP	Apl2/β1
	Apl2-2mCherry	
	Apm1-GFP	Apm1/μ1
	Apm1-2mCherry	
AP-2	Apl1-2GFP	Apl1/β2
	Apl1-2mCherry	
	Aps2-2GFP	Aps2/σ2
	Aps2-2mCherry	
AP-3	Apl5-GFP	Apl5/δ
	Apl5-2mCherry	
	Apm3-GFP	Apm3/μ3
	Apm3-2mCherry	
retromer	Vps26-GFP	Vps26
	Vps26-2mCherry	
	Vps5-GFP	Vps5
	Vps5-2mCherry	

**Table 2 membranes-15-00209-t002:** Location and potential function of six coat protein complexes in yeast.

Coat	Location Reported	Location Observed in This Study	Potential Function
COPII	ER, early Golgi [23]	ER, early Golgi, transport vesicles (possible)	Transport from the ER to cis-Golgi [23]
COPI	Early Golgi [25]	Early Golgi, late Golgi	Transport from the Golgi to ER, transport between Golgi stacks [7]; essential for Golgi cisternal maturation and dynamics [24,25]
AP-1	Late Golgi [27]	Late Golgi, transport vesicles	Transport from the late Golgi to endosome [10]; essential for Golgi cisternal maturation [27]
AP-2	Plasma membrane [30], bud neck [29]	Plasma membrane, bud neck	Transport from plasma membrane to endosome [29,30]
AP-3	Late Golgi [10,33], vacuole [34,35]	Late Golgi, vacuole, transport vesicles	Transport from the late Golgi to vacuole [10,33]
retromer	Late endosome [38], vacuole [37,38]	Late endosome, vacuole; transport vesicles (possible)	Transport from endosome to Golgi [8]

## Data Availability

The original contributions presented in this study are included in the article. Further inquiries can be directed to the corresponding authors.

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
