# Peer review of "A Set of Fluorescent Protein-Based Markers for Major Vesicle Coat Proteins in Yeast"

_membranes, 2025, doi:10.3390/membranes15070209_

Round 1
Reviewer 1 Report
Comments and Suggestions for Authors
Cui et al. have constructed a set of yeast strains expressing different vesicle coat subunits, tagged with a fluorescent protein at the C-terminus. They have tagged two different subunits of the coat complexes COPII, COPI and retromer, and two subunits of the clathrin-adaptor complexes AP-1, AP-2 and AP-3. They then use wide-field fluorescent microscopy to analyze the localization of these subunits in yeast, and analyze their colocalization with a previously-established set of fluorescent organelle markers (ER, early and late Golgi, late endosome, vacuole, plasma membrane and mitochondria).
The main problem of this manuscript is the lack of any new insight; for this reason, it cannot be published as a research paper.
If the aim of this manuscript is to report on a new set of research tools, it should be noted that these tools are not particularly novel either; rather, similar genetic tools have already been published by many groups in many different studies and in principal should all be available upon request (or through Addgene). For example, in 2003, Huh, et al., Nature 425, 686-691 (2003) published a genome-wide set of GFP-tagged strains, where all yeast genes were systematically tagged at their C-termini, and the localization of the resulting fusion proteins was analyzed in an automated manner, including using colocalization with organelle markers; the results of this work can be accessed via the website https://yeastgfp.yeastgenome.org/. This study was followed by many other genome-wide localization studies; none are cited in the present work.
As far as I can tell, the analysis of Cui et al largely agrees with the results of Huh 2003, obtained 22 years ago. For some of the subunits in question, the signal was too weak, therefore Cui et al used a tandem mCherry or GFP tag. However, they do not verify whether the tag affects the activity of the protein in question; this should absolutely be verified for these genetic tools to be useful for other researchers.
Another problem is that the authors tend to over-interpret their data, making statements that are not supported by their data (because they do not have sufficient resolution or signal-to-noise ratio). For example, in the Discussion, lines 362-382 should be deleted; it is not possible to separate signals on vesicles from donor/acceptor compartments using single images and conventional fluorescent microscopy. At the least, the authors should perform video microscopy and single-particle tracking with a confocal and/or super-resolution microscope.
A vast body of literature is available on the localization and function of different vesicle coat proteins. While I agree with the authors that it is of interest to analyze these coat proteins all together, the present work unfortunately doesn't bring any novel insight and would be more useful as a literature review (with a significantly expanded reference list). It cannot be published as an original research study.
Author Response
Comments 1: [Cui et al. have constructed a set of yeast strains expressing different vesicle coat subunits, tagged with a fluorescent protein at the C-terminus. They have tagged two different subunits of the coat complexes COPII, COPI and retromer, and two subunits of the clathrin-adaptor complexes AP-1, AP-2 and AP-3. They then use wide-field fluorescent microscopy to analyze the localization of these subunits in yeast, and analyze their colocalization with a previously-established set of fluorescent organelle markers (ER, early and late Golgi, late endosome, vacuole, plasma membrane and mitochondria).
The main problem of this manuscript is the lack of any new insight; for this reason, it cannot be published as a research paper.
If the aim of this manuscript is to report on a new set of research tools, it should be noted that these tools are not particularly novel either; rather, similar genetic tools have already been published by many groups in many different studies and in principal should all be available upon request (or through Addgene). For example, in 2003, Huh, et al., Nature 425, 686-691 (2003) published a genome-wide set of GFP-tagged strains, where all yeast genes were systematically tagged at their C-termini, and the localization of the resulting fusion proteins was analyzed in an automated manner, including using colocalization with organelle markers; the results of this work can be accessed via the website https://yeastgfp.yeastgenome.org/. This study was followed by many other genome-wide localization studies; none are cited in the present work.
As far as I can tell, the analysis of Cui et al largely agrees with the results of Huh 2003, obtained 22 years ago. For some of the subunits in question, the signal was too weak, therefore Cui et al used a tandem mCherry or GFP tag. However, they do not verify whether the tag affects the activity of the protein in question; this should absolutely be verified for these genetic tools to be useful for other researchers.]
Response 1: [We understand you have concerns over the utility of small scale tools when genome scale tools are available. Thank you for reminding us of the GFP library. We have it in our freezer, and use it when a project calls for it. However we would like to point out that each types of tools have their own use cases.
We have created similar plasmid tools in the past for organelles, Atg proteins and others, and deposited them in Addgene. To this day, Addgene has distributed 548 samples to more than a hundred research groups. We can see from citation records that people are using our tools in their publications.
Regarding protein function, this is an important aspect. Following your suggestion, we verified the functionality of our coat chimera constructs using a collection of assays, and found that our constructs were functional. These new data are presented in new Fig. 2- 5.]
Comments 2: [Another problem is that the authors tend to over-interpret their data, making statements that are not supported by their data (because they do not have sufficient resolution or signal-to-noise ratio). For example, in the Discussion, lines 362-382 should be deleted; it is not possible to separate signals on vesicles from donor/acceptor compartments using single images and conventional fluorescent microscopy. At the least, the authors should perform video microscopy and single-particle tracking with a confocal and/or super-resolution microscope.
A vast body of literature is available on the localization and function of different vesicle coat proteins. While I agree with the authors that it is of interest to analyze these coat proteins all together, the present work unfortunately doesn't bring any novel insight and would be more useful as a literature review (with a significantly expanded reference list). It cannot be published as an original research study.]
Response 2: [We understand that wide field light microscopy has a resolution limit of about 250nm. In cases that when two structures do not co-localize, it is reasonable to conclude that the two are at least 250 nm apart, which is more than the usual size of most transport vesicles in yeast. Therefore when we observed that half of AP-1 puncta were away from their known donor/acceptor compartments by light microscopy, the conclusion that half of AP-1 puncta are not on their donor/acceptor compartments is justified. For other coats, each has its unique limitations that prevented us from reaching such type of conclusions, and we outlined these limitations in the discussion.
To our knowledge, the present work is the only one that provided colocalization quantifications of coat proteins against a comprehensive set of organelle markers. If we missed a similar work, please kindly notify us so that we can compare our quantification result with an existing one. We have cited studies that presented imaging data on coat complexes and organelle markers.]
Reviewer 2 Report
Comments and Suggestions for Authors
The manuscript by Cui et al presents a systematic and comprehensive study of the localization of vesicle coat proteins in yeast at endogenous expression levels. The work presented is well-performed, sound and interesting for the field. The major limitation of this work is that there is no evidence presented that the integrated constructs are correctly expressed, which could be verified my immunoblot using anti-fluorescence protein antibodies, or that the fusions are functional, by performing phenotypic studies on the integrated strains. The authors rely on fluorescence as proof of expression and perform a valuable set of experiments. The article is worth publishing with minor revisions if the editors consider that parallel proof of expression and function of the integrated strains is dispensable. Journals with a strong biochemistry or molecular biology orientation would demand this, whereas those focused exclusively in cell biology outside the top profile might not.
Minor points.
It is stated that cells are resuspended in water prior to microscopic observation. I am worried that this may cause a hypoosmotic shock, temporarily disrupting some vesicular trafficking pathways and leading to artefactual conclusions, accumulating vesicles at docking sites or reorganizing secretory pathways to face the osmotic challenge.
Due to the large amount of images in each panel, the size of individual images is small. This is OK for those that are meant for co-localizations, as the authors offer an interpretation. But in the case of Fig. 1 it is very hard to visually support any of the statements in the text with the images. Negative black and white images (black over white) would be easier for the reader to interpret here, as the color is not crucial for interpretation in this particular figure.
Statistical significance is not provided in Fig.4, which may be especially important for Anp1-mCherry co-localization.
In Figs. 9 and 10, please explain why two fields are presented, in contrast with other figures. This happens also in Fig. 4, first two rows, because I assume they correspond to peripheric vs. perinuclear ER (please explain better), but the rationale supporting why two images are shown for each experiment in these particular figures is not clear to me.
The article is generally well written, but please have it checked by a native speaker. The use of plurals is often wrong. Line 10: transport interconnects; line 102: two groups; 106, scheme (…) is; 109, set (…) is; 226, network is; 343, puncta represent;
Check the sentence in lines 38-40. The yeast Saccharomyces cerevisiae is not a “study”
Line 143 Choose should be chose
Line 232. The verb is missing. “they were substantially…”?
Line 258. Person should be Pearson
Lines 265, 288 etc. as fig. 3 should be “as in Fig.3” or “as those in Fig.3”
Line 275. Addition should be additional.
Headings on Table 2. “Location observed of the paper” does not make much sense. “Location observed here”?
Make sure all species and gene names are in italics. See line 39, 124, etc
CLH plasmids are Clh in the Table, please homogenize.
Please revise thoroughly the supplementary table: mNeoGreen should be mNeonGreen. HandIII should be HindIII, I guess. When citing genes please do not use protein nomenclature (applies to all the column: e.g. Cop1 should be COP1 (in italics). Homogenize typo.
Author Response
Comments 1: [The manuscript by Cui et al presents a systematic and comprehensive study of the localization of vesicle coat proteins in yeast at endogenous expression levels. The work presented is well-performed, sound and interesting for the field. The major limitation of this work is that there is no evidence presented that the integrated constructs are correctly expressed, which could be verified my immunoblot using anti-fluorescence protein antibodies, or that the fusions are functional, by performing phenotypic studies on the integrated strains. The authors rely on fluorescence as proof of expression and perform a valuable set of experiments. The article is worth publishing with minor revisions if the editors consider that parallel proof of expression and function of the integrated strains is dispensable. Journals with a strong biochemistry or molecular biology orientation would demand this, whereas those focused exclusively in cell biology outside the top profile might not.]
Response 1: [Thank you for the critical evaluation of our work. Following your suggestion, we verified the expression of the coat constructs by immunoblotting, and verified the functionality of each coat using various assays. The results are presented in Fig. A1, Fig.2-5, and text additions are highlighted. ]
Comments 2: [Minor points.
It is stated that cells are resuspended in water prior to microscopic observation. I am worried that this may cause a hypoosmotic shock, temporarily disrupting some vesicular trafficking pathways and leading to artefactual conclusions, accumulating vesicles at docking sites or reorganizing secretory pathways to face the osmotic challenge.]
Response 2: [Thank you for attention to technical perfection. We did a test to see how long a hypoosmotic shock would have a noticeable impact on vacuolar morphology (i.e., fusion to generate large vacuoles), and found that it took 10 min or more. During all our previous experiments, the centrifuge was placed nearby the microscope. And every time a single sample was resuspended and imaged. So it took less than 5 minutes from the resuspension to the actual imaging, which was shorter than the time needed for a visible effect. Nevertheless, as a precaution, we added a sentence in our method section to alert readers of this potential. The sentence states: ”note that suspension in water can lead to hypoosmotic shock”.]
Comments 3: [Due to the large amount of images in each panel, the size of individual images is small. This is OK for those that are meant for co-localizations, as the authors offer an interpretation. But in the case of Fig. 1 it is very hard to visually support any of the statements in the text with the images. Negative black and white images (black over white) would be easier for the reader to interpret here, as the color is not crucial for interpretation in this particular figure.
Response 3: [Thank you for the suggestion. We added inverted black on white versions of the fluorescent micrographs in Fig.1 as Fig. A2.]
Comments 4: [Statistical significance is not provided in Fig.4, which may be especially important for Anp1-mCherry co-localization.
Response 4: [We performed t-test comparing the early Golgi ratios against other sites with residual colocalization. The colocalization of Sec23 with Anp1 (early Golgi) was significantly higher than with Sec7 (late Golgi/early endosomes) or Vps4 (late endosomes). We added the information into the figure and figure legends (Fig. 7&8 in revised manuscript).]
Comments 5: [In Figs. 9 and 10, please explain why two fields are presented, in contrast with other figures. This happens also in Fig. 4, first two rows, because I assume they correspond to peripheric vs. perinuclear ER (please explain better), but the rationale supporting why two images are shown for each experiment in these particular figures is not clear to me.
Response 5: [Following your comments, we included additional explanation in the figure legends (Fig. 13&14 in revised manuscript). AP-2 subunits existed in two very different locations, one containing a large amount of protein at the bud necks, the other small puncta scattered along the PM. The reason for including two zoomed in areas was to present the situation in both locations. The situation on COPII and ER was similar, as we were demonstrating that COPII puncta were present on both peripheral and nuclear ER.]
Comments 6: [The article is generally well written, but please have it checked by a native speaker. The use of plurals is often wrong. Line 10: transport interconnects; line 102: two groups; 106, scheme (…) is; 109, set (…) is; 226, network is; 343, puncta represent;
Check the sentence in lines 38-40. The yeast Saccharomyces cerevisiae is not a “study”
Line 143 Choose should be chose
Line 232. The verb is missing. “they were substantially…”?
Line 258. Person should be Pearson
Lines 265, 288 etc. as fig. 3 should be “as in Fig.3” or “as those in Fig.3”
Line 275. Addition should be additional.
Headings on Table 2. “Location observed of the paper” does not make much sense. “Location observed here”?
Make sure all species and gene names are in italics. See line 39, 124, etc
CLH plasmids are Clh in the Table, please homogenize.
Please revise thoroughly the supplementary table: mNeoGreen should be mNeonGreen. HandIII should be HindIII, I guess. When citing genes please do not use protein nomenclature (applies to all the column: e.g. Cop1 should be COP1 (in italics). Homogenize typo.]
Response 6: [Thank you for catching the text issues. All have been revised. Note that in the plasmid table, promoters are denoted using gene name, and ORFs are denoted by protein name. An ORF without a promoter name indicates that the chimera construct does not have a promoter. ]
Reviewer 3 Report
Comments and Suggestions for Authors
The manuscript Cui et al. “A set of fluorescent protein based markers for major vesicle coat proteins in yeast” is of a special interest. The correct and well interpreted methods of intracellular organelles visualization are in demand by modern studies focusing on plasticity of the endomembrane system. Authors designed set of markers (six coat protein complexes) allowing live cell imaging of ER, Golgi and plasma membrane vesicles.
Introduction is well planned and gives full required information about modern state of knowledge in the field of the importance of coat proteins subcellular localization. Material and methods are presented in an accessible manner and fully described used approaches. Results are well illustrated and further discussed.
Taken together I can conclude that the manuscript subject and obtained data fit the journal scope, and in present form it can be recommended for publication.
Author Response
Comments 1: [The manuscript Cui et al. “A set of fluorescent protein based markers for major vesicle coat proteins in yeast” is of a special interest. The correct and well interpreted methods of intracellular organelles visualization are in demand by modern studies focusing on plasticity of the endomembrane system. Authors designed set of markers (six coat protein complexes) allowing live cell imaging of ER, Golgi and plasma membrane vesicles.
Introduction is well planned and gives full required information about modern state of knowledge in the field of the importance of coat proteins subcellular localization. Material and methods are presented in an accessible manner and fully described used approaches. Results are well illustrated and further discussed.
Taken together I can conclude that the manuscript subject and obtained data fit the journal scope, and in present form it can be recommended for publication.]
Response 1: [Thank you for reviewing and appreciating our work.]